# Broadband High-Reflection Dielectric PVD Coating with Low Stress and High Adhesion on PMMA

**Zizheng Li [1], Qiang Li [1,\*], Xiangqian Quan [2,\*], Xin Zhang [1], Chi Song [1], Haigui Yang [1], Xiaoyi Wang [1] and Jinsong Gao [1]**

1    Key Laboratory of Optical System Advanced Manufacturing Technology, Changchun Institute of Optics, Fine Mechanics and Physics, Chinese Academy of Sciences, Changchun 130033, China; lizizheng@ciomp.ac.cn (Z.L.); Zhangxin@ciomp.ac.cn (X.Z.); songchi@ciomp.ac.cn (C.S.); yanghg@ciomp.ac.cn (H.Y.); wangxy@ciomp.ac.cn (X.W.); gaojs@ciomp.ac.cn (J.G.)
2    Institute of Deep-sea Science and Engineering, Chinese Academy of Sciences, Sanya 572000, China
\*    Correspondence: liqiang@ciomp.ac.cn (Q.L.); quanxq@idsse.ac.cn (X.Q.)

**Abstract:** Polymethyl methacrylate (PMMA) is an attractive optical plastic that is widely used in augmented reality, virtual reality devices, display, wearable devices, portable optical equipment, and lightweight optics. Thin film prepared by physical vapor deposition (PVD) is the primary choice of coating on PMMA. However, it faces problems with coating adhesion and stress. In this paper, we analyze the problems existing in the current PMMA high-reflection (HR) coating in detail and propose a way to effectively solve issues with bonding force and stress. Based on the current research background, the bonding force was enhanced by introducing a special hard coating as the connection layer between the dielectric film and the substrate. After comparing the stresses of different coating materials and material combinations, the optimal combination of $Nb_2O_5$ and $SiO_2$ was determined, and the requirements were successfully prepared. An HR coating that satisfies requirements, with low stress and excellent environmental adaptability, was successfully prepared. Based on this, a broadband HR coating from 750 to 1550 nm was formed on the surface of PMMA by adjusting the partial pressure of oxygen.

**Keywords:** PMMA; high-reflection coating; coating stress; coating adhesion

## 1. Introduction

Polymethyl methacrylate (PMMA) is a kind of transparent thermoplastic (also known as acrylic or acrylic glass) which is often used in sheet form as an economic, lightweight, or shatter-resistant alternative to optical glass and polycarbonate (PC) [1–4], especially when the properties of tensile strength, flexural strength, transparency, polish ability, and UV tolerance are more important. In addition, PMMA does not contain the potentially harmful subunits found in PC. It is usually preferred because of its moderate properties, excellent visible and infrared transmittance, easy handling, simple processing, and low cost [5,6]. In the field of optics, it is getting more and more attention from designers. PMMA optical plastic can also replace ordinary optical glass. For example, for its easy-to-form feature, it can be made into a spherical mirror or a lens with different optical thin film to correct aberrations and improve image quality [7,8]. In fabricating the PMMA optical substrate, thermal injection molding is the most common process. The surface precision of the mold determines the surface accuracy of PMMA. For improving the conditions of the thermal injection molding process, PMMA could be used directly as an optical substrate without traditional optical precision manufacturing. In our work, PMMA has mainly been used as a coating substrate and, in other situations, it can also be used as a coating material [9–11].

Owing to the advantages of PMMA, coatings on it are widely used in optical systems. Thus far, it continues to be an important research topic, and the focus is the thin film design and coating process. Nowadays, augmented reality, virtual reality devices, display, wearable devices, portable optical equipment, and lightweight optics have a strong demand for coatings on PMMA [12–15]. Generally, optical thin film that needs to be prepared on the surface of PMMA in an optical system includes antireflection coating, spectroscopic coating, and high-reflection coating. Thin film prepared by physical vapor deposition (PVD) is the leading choice of coatings on PMMA due to its potential for mass production, low cost, and productive efficiency. However, it currently still faces several challenges [16–18]. For our main purpose of visible or near infrared images and observations, the problems mainly concentrate on the coating stress, adhesion between coating and PMMA, and environmental resistance. PMMA optical substrates are prone to internal stress during molding and are not easily eliminated. When the internal stress is large, after the film material is deposited on the surface of the substrate, the internal stress is transferred to the film layer, thereby greatly reducing the stability of the film layer and even cracking and wrinkling it [19,20]. Therefore, measures must be taken in order to minimize internal stress and, at the same time, the coated layer should be able to withstand as much internal stress as possible, without cracking or falling off. Additionally, the optical plastic surface has a high-impedance characteristic, and it is easy to generate static electricity after rubbing. Since PMMA has poor heat resistance and high water absorption, it is critical to set vacuum coating parameters and to improve film adhesion. In addition, if the difference in the thermal expansion coefficient between PMMA and thin film materials is too large, stress will be generated due to temperature changes during or after film formation and film layer peeling occurs.

At present, coatings with less thickness could be successfully deposited onto PMMA. Less thickness means a lower probability of stress. Many ways have been suggested to increase adhesion, such as annealing, adding a special connection layer, or using a hard coating. Adhesion improving processes are always carried out before optical thin film deposition. In our work, we also performed similar hardening by introducing superior silicon-based organic solvents. Thus, the most critical issue of coatings on PMMA is thick film, especially the HR PVD dielectric coating. Our investigations are mainly about solving the stress and adhesion of PVD coatings on PMMA. By adopting a special bonding layer material and substrate cleaning process, adhesion was significantly improved. By carefully choosing deposition materials and circularly revising deposition parameters, stress problems of HR coating on PMMA can be perfectly solved. Additionally, we put forward a new method of oxygen deficit deposition to realize partially visible to near infrared broadband HR coating.

## 2. Materials and Methods

Before coating deposition, PMMA substrates were carefully cleaned and underwent annealing treatment to enhance adhesion. The dimensions of the PMMA substrates are 50 mm × 50 mm × 3 mm. All of the chemicals used were of reagent grade or better. Firstly, the PMMA substrates were immersed in a solution of sodium hydroxide (60 g/L), sodium carbonate (20 g/L), and sodium phosphate (30 g/L) at 30–70 °C for 20–30 min. Then, they were sequentially rinsed with deionized water 2 to 3 times. The water was allowed to volatilize at room temperature and was heated to about 80 °C for 2 h to reduce the internal stress during the PMMA substrate molding process and the absorbed moisture.

The deposition processes were all realized in an OPTRUN (OTFC-1300, OPTRUN, Kawagoe, Japan) coating machine, and the deposition method was ion beam-assisted electron beam evaporation. The thickness of each layer was estimated using the data provided by the reflected optical value. The deposition rate of the coating materials was controlled using the crystal oscillate system. The low refractive index material of HR coating was $SiO_2$, while the high refractive index material was selected from $TiO_2$, $Ta_2O_5$, $ZrO_2$, $HfO_2$, and $Nb_2O_5$ through experimental comparison. Additionally, we evaluated the coating thickness, coating surface shape, stress, inner structure, surface reflectance, surface morphology, and hardness using a stylus profiler (Nanomap 500LS, CAEP, Santa Clara, CA, USA), digital wavefront interferometer (Mark III-GPI, Zygo Corporation, Berwyn, PA, USA),

stress measurement instrument (MOS Ultra-Scan, K-Space Associates, Inc., Dexter, MI, USA), scanning electron microscope (SEM, JSM-6510, JEOL, Tokyo, Japan), spectrophotometer (Lambda 1100, PerkinElmer, Waltham, MA, USA), atomic force microscope (AFM, EDG, Bruker, Billerica, MA, USA), and nano indentation (Nano Indenter G200, Agilent, Santa Clara, CA, USA), respectively.

## 3. Results and Discussion

The PMMA substrate has good transparency in the visible spectrum. Compared with PC and optical glass, PMMA has its disadvantages, such as impact strength, chemical resistance, and heat resistance. At the same time, it introduces more coating preparation problems. In the initial stages of our experiment, AR (antireflection)coating on PMMA was realized by e-beam evaporation. The average transmittance from 400 to 720 nm reaches 98.8% owing to two-sided AR coatings, as shown in Figure 1. The absorbance of the PMMA substrate in the visible spectrum is less than 0.1%. However, when the total thickness of the coating is larger or the number of coating layers is greater, the deposition process is no longer that simple. Problems focus on the adhesion between the coating and substrate, as well as the coating stress.

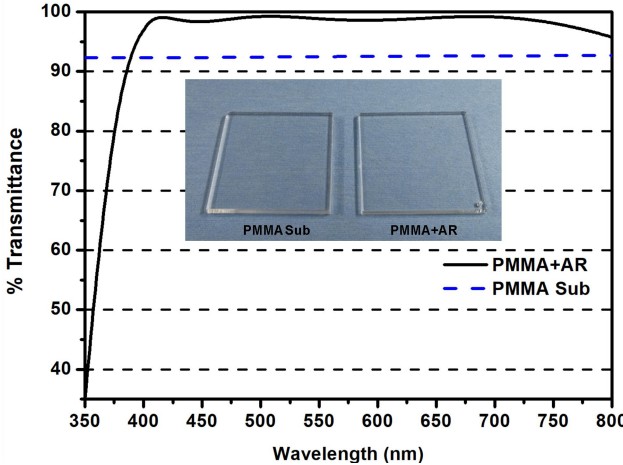

**Figure 1.** Transmittance of Polymethyl methacrylate (PMMA) substrate (blue line) and PMMA with two-side AR coatings (black line).

The deposition processes were all realized in a vacuum chamber (OTFC-1300, OPTRUN, Kawagoe, Japan). To enhance the adhesion between the PMMA substrate and multi-layer coating, we put forward a transparent methyl silicone solvent as the connection layer. Generally, this kind of treatment can be collectively referred to as a PMMA hardening treatment. To overcome the weak scratch resistance of PMMA, many kinds of hard-coating (HC) media, such as melamine-, acrylic-, and urethane-based chemicals, have been developed and used. For our HC, with a different formula, it had a different hardness after heating at 80 °C The largest hardness obtained was 0.26 GPa, tested using a K-space instrument, as shown in Figure 2a. We used the spin coating method at 2000 rpm with considerations for thickness uniformity. The transmittance curve of a single side hard coating on PMMA was also measured, as shown in Figure 2b. From this data, its refractive index can be calculated, which is 1.56 times larger than that of the PMMA. The physical thickness of HC coating is almost 1.05 μm, which can be calculated using a transmittance curve or directly measured using a stylus profiler. The thickness results obtained by the two methods were nearly the same. The thickness uniformity error was less than ±1% from the center to the edge of PMMA sample. When the new refractive index material HC was introduced, the coating design needed to be revised slightly. The coating with HC could withstand a repeated pull of CT-24 tape.

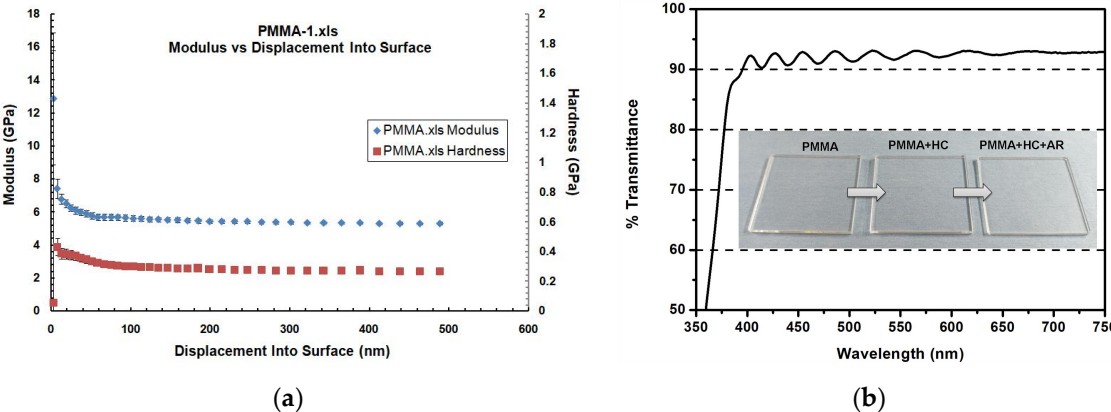

**Figure 2.** (**a**) Hardness and modulus of the hard coating obtained from nano intender; (**b**) Transmittance of PMMA with HC and two-side AR coatings.

The first broadband HR coating studied was a typical application in virtual reality or augmented reality devices. The reflective spectrum was 400–700 nm, and the incident angle was 0°–30°. The thin film formula is very simple, which can be expressed as PMMA Substrate/$(HL)^5$ $1.3(HL)^5 1.7(HL)^5$/Air. *H* and *L* represent the high and low refractive index deposition material, respectively. Using a simplex optimization process, we could get a smoother reflectance curve. When *H* was $TiO_2$ and *L* was $SiO_2$, we got a design result as shown in Figure 3. The three lines with different colors represent the reflectance when the incident angle was 0°, 15°, and 30°. The average reflectance at different incident angles was greater than 96.3%. If the *H* and *L* materials are changed, the thin film only needs to get re-optimized to get roughly the same reflectance result.

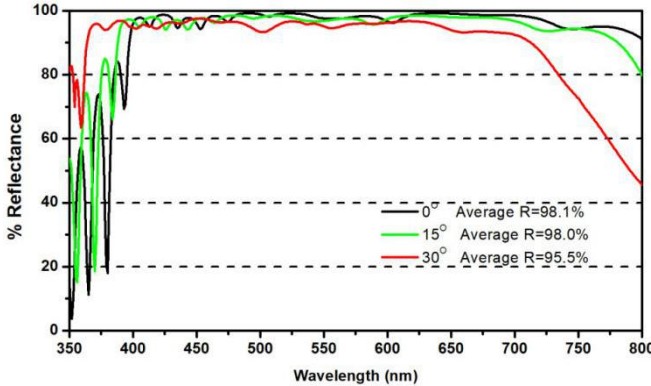

**Figure 3.** Calculated reflectance of designed 30 layers high-reflection (HR) coating at incident angles of 0° (black line), 15° (green line), and 30° (red line).

For the design of the coatings on PMMA, there were three concepts explored in our study. The high refractive index material must be carefully selected to reduce the stress property. Silicon oxide was preferred as a low refractive index material for its stability and process repeatability. Minimum thickness and layer number leads to the shortest deposition time and temperature rise. For choosing the *H* material of the broadband HR coating, two series of experiments are carried out. First, we prepared single-layer coating samples of different materials, including $SiO_2$, $TiO_2$, $Ta_2O_5$, $ZrO_2$, $HfO_2$, and $Nb_2O_5$, as shown in Figure 4a. The coating substrates were φ50 mm × 3 mm PMMA. For processes using different materials, deposition parameters were kept as consistent as possible, including deposition rate, heating temperature, ion source power, sample spatial location, and oxygen partial pressure. Evaporation rates were all 0.3 nm/s; the background vacuum was $9.0 \times 10^{-4}$ Pa; the heating temperature was 220 °C; the voltage and current of the RF (radio frequency) ion source were 1100 V and 1000 mA, respectively; the sample was 1.3 m in height above the electronic gun; coating

vacuum after oxygenation was $2.0 \times 10^{-2}$ Pa; and the coating thickness was 300 nm. By measuring the changes in the substrate curvature before and after coating, the coating stress could be calculated by using the Stoney equation below [21–23].

$$\sigma_{\mathrm{f}} = \frac{E_{\mathrm{s}}}{1 - V_{\mathrm{s}}} \cdot \frac{t_{\mathrm{s}}^2}{6 R t_{\mathrm{f}}} \qquad (1)$$

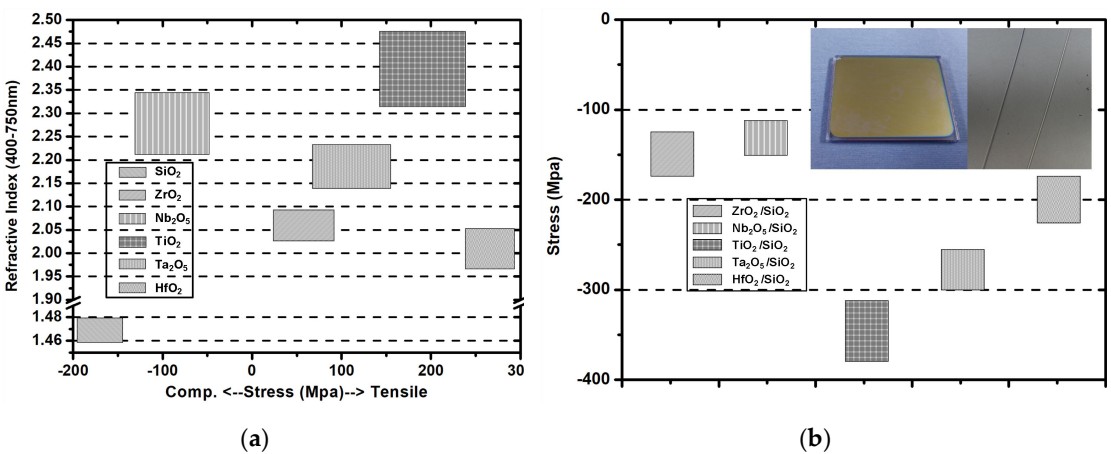

**Figure 4.** (**a**) Refractive index of different coating materials, including $SiO_2$, $TiO_2$, $Ta_2O_5$, $ZrO_2$, $HfO_2$, and $Nb_2O_5$; (**b**) Stress test results of 30 layers HR coating samples of different high and low refractive index material combination, image in the plot exhibit the rupture of HR coating caused by large coating stress.

$E_{\mathrm{s}}$ and $V_{\mathrm{s}}$ are the modulus of elasticity and Poisson's ratio of the PMMA substrate, respectively. They are obtained using the hardness test of PMMA. Besides, $t_{\mathrm{s}}$ and $t_{\mathrm{f}}$ are the thickness of the substrate and coating, respectively. $R$ is the radius of the curvature of the PMMA substrate. Through the measurement and calculation of different samples, we got the data of the refractive index and coating stress, as shown in the Figure 4a. Stress changing was caused by multiple test measurement results.

In the same way, 30-layer HR coating samples of different $H$ materials were prepared. The results are shown in Figure 4b. The biggest challenge for HR coating on PMMA is the stress problem. When the deposition material and process parameter were not fine enough, coatings always exhibited the crack or fall off phenomenon. Many test samples showed a cracking of the film layer. As the micrograph shows in Figure 4b, although the HR coating layer did not fall off directly, many cracks inside the coating appeared. Unlike single-layer films, the stress development of multilayer films involves the combination of thermal parameters of various thin film materials and the rational matching of multiple interfaces, which is a more complicated process. Through a comprehensive evaluation of the HR coating samples, it was found that the $Nb_2O_5$ and $SiO_2$ combination has a better performance with regards to a lower probability of film rupture. Additionally, it should be noted that the coating stress result of different samples is a range but not a constant value in Figure 4. This was mainly caused by the errors measured in multiple experiments and the temperature differences of the film formation area of different samples.

The problem of coating stress on PMMA will invalidate many previous methods. PMMA can only withstand temperatures up to 80 °C and cannot be subjected to high temperature annealing to remove residual stress. In addition, the temperature was controlled as much as possible during the coating process, and the electron beam or ion beam themselves had a heating effect. Therefore, at the beginning of the experiment, the heating method was not used and the coating was started at room temperature. It should also be noted that, since the coating process was a continuous heating process,

the effect of temperature change on the refractive index and stress accumulation effect of the material should be considered.

After determining the selected coating material by adjusting the material evaporation rate and ion beam source parameter, we obtained the optimized deposition parameters. The highest deposition temperature was controlled to be less than 80 °C. Finally, the HR coating on the PMMA with no obvious defect was prepared successfully, and the process repeatability and stability were very good. The measured reflectance data is shown in Figure 5a. The average reflectance is 95.5% at 30°, which has a 0.8% reduction compared with the design. This is explained by the fact that the deposition process was started at room temperature but the whole process temperature kept rising. The refractive index of the deposition material was not steady and some absorption also led to reflectance reduction. Based on interferometer measurement, the surface roughness RMS (Root Mean Square) of this as-deposited HR coating sample was less than $1/20\lambda$, as shown in Figure 5b.

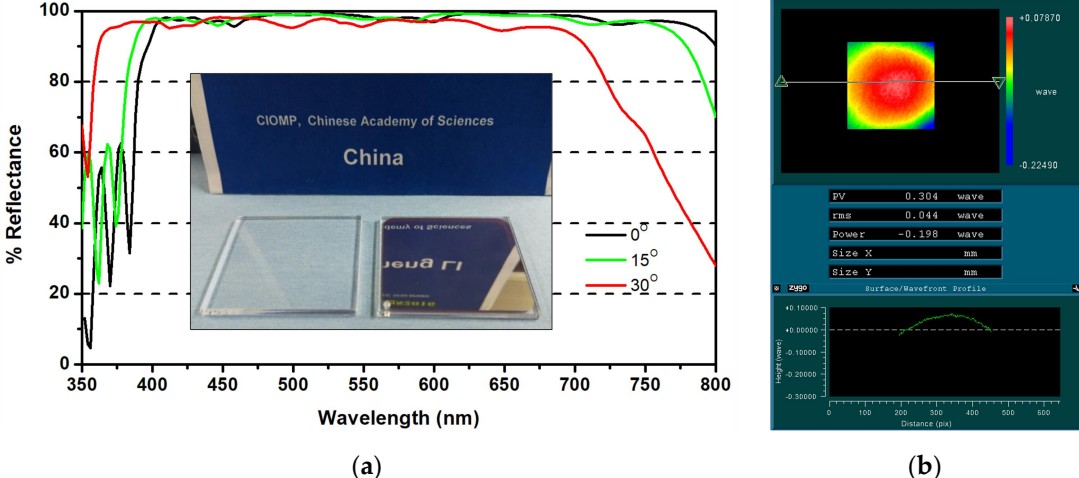

**Figure 5.** (**a**) Measured reflectance of the optimized 30 layers HR coating and samples image (left is PMMA substrate, right is HR coating sample); (**b**) Surface shape test result measured by the interferometer.

We also conducted an environmental durability test. The test was carried out using dry heat at 80 °C, cold at −20 °C, and heat for 72 h at 80 °C and 80% relative humidity. In comparing the samples before and after the test, there is no obvious difference in outlook and reflectance, as shown in Figure 6a. On the other hand, we used K-space test equipment to measure the stress distribution of samples with different deposition rates. The lowest average stress measured was −85 MPa at a 0.2 nm/s deposition rate, as shown in Figure 6b. When deposition rates rose to 0.3 nm/s, the average stress of the HR coating increased to −165 MPa. Samples with different deposition rates also experienced environmental testing, and the deposition rate did not affect coating durability.

As we have been emphasizing, multi-layer coating stress is a complex model, which can be significantly influenced by lots of deposition parameters. Besides the deposition rate, temperature, spatial structure size, and oxygen partial pressure are likewise key parameters. We found that with different oxygen partial pressures, the composition of niobium oxide will be obviously changed. This will directly impact the extinction coefficient ($k$) of niobium oxide, light absorption, and coating stress. In Figure 7a, we can see the strong dependence of the extinction coefficient on oxygen partial pressure. From the comparison of the 30-layer HR coating samples of $Nb_2O_5$ and $Nb_2O_x$, both have excellent surface appearance and film quality. The measured reflectance of HR coating deposited at $1.0 \times 10^{-3}$ Pa can be observed as the red line in Figure 7b. Its coating stress distribution and surface shape marked as the $Nb_2O_x$-1# are shown on the top, with mean values of −32 MPa and $0.018\lambda$, respectively. For the regrettable absorption of niobium oxide, the averaged reflectance from 400–700 nm had a 4% reduction compared with the HR coating mentioned before, as demonstrated by the

black line in Figure 7b. Based on the method of regulating oxygen partial pressure, we successfully prepared a broadband HR coating, which has an average 98.5% reflectance from 750 to 1550 nm, as the green line shows in Figure 7b. The oxygen partial pressure was set at $0.5 \times 10^{-3}$ Pa. The coating stress distribution and surface shape marked as the $Nb_2O_x$-2# are shown below, with mean values of $-45$ MPa and $0.04\lambda$, respectively. The total thickness of this broadband HR coating was 7.7 μm. For such a thick coating on PMMA, the sample showed no obvious appearance defect and good environmental durability. After being repeatedly pulled by tape CT-24, the broadband HR coating exhibited a good adhesion performance.

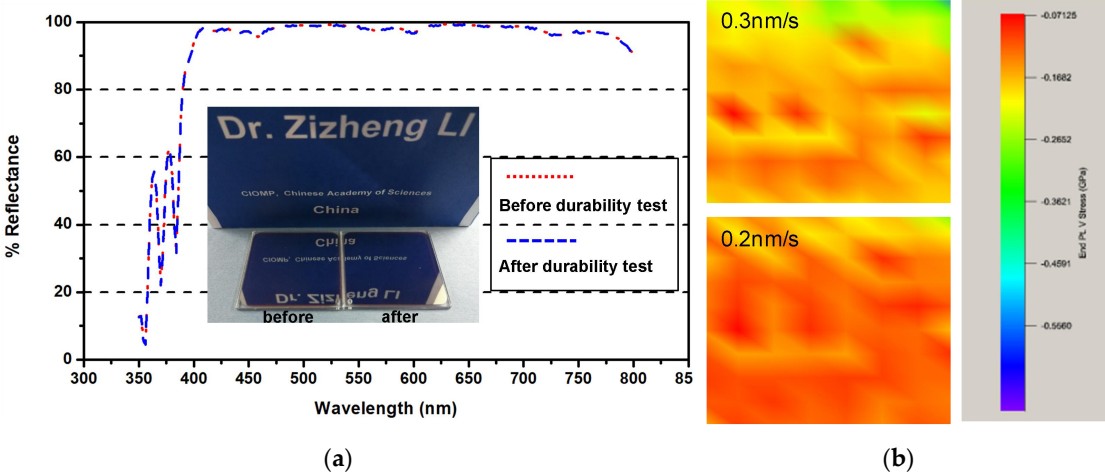

**Figure 6.** (**a**) Measured reflectance of HR coating before (red line) and after (blue line) the durability test; (**b**) Stress test results of PMMA samples with different deposition rates directly obtained from the MOS Ultra-Scan of K-space.

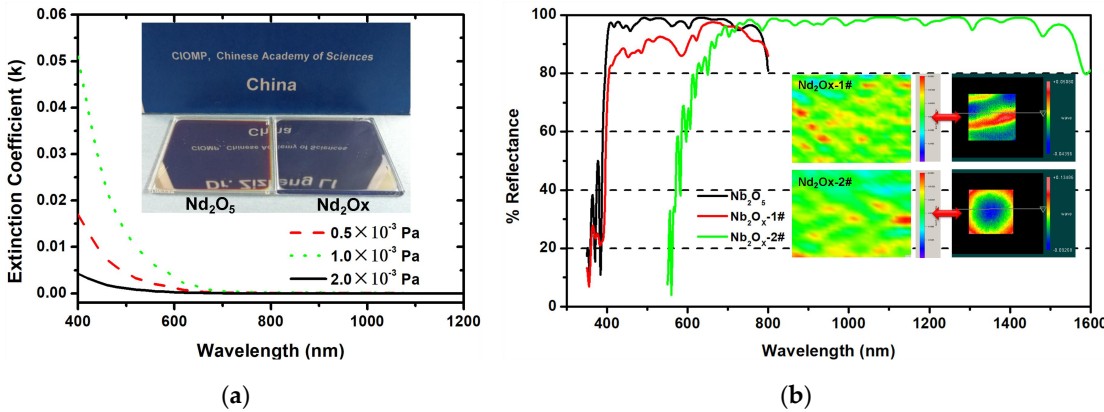

**Figure 7.** (**a**) The dependence of the extinction coefficient on oxygen partial pressure; (**b**) Reflectance, coating stress, and surface shape of HR coating samples with different oxygen partial pressure.

## 4. Conclusions

In summary, after analyzing the difficulty of plating the reflective film on the PMMA, the problem of the bonding force between the PMMA and the dielectric film was solved by introducing a special ratio of the HC connecting layer, which can withstand repeated pulling of the CT-24 tape. The maximum hardness of HC is 0.26 GPa. The film stress of different materials and material combinations were compared through a series of experiments, and $Nb_2O_5$ and $SiO_2$ were finally determined to be the best combination. Subsequently, a 400–700 nm, 0°–30° broadband HR coating was prepared. The film layer was free from cracks, the surface quality was good, and the reflective surface shape was better than $1/20\lambda$. In addition, the environmental reliability test of the prepared

reflective film was carried out, and the influence of deposition rate on film stress was analyzed. Finally, the method of ingeniously adjusting the oxygen partial pressure was proposed to further reduce the film stress of the reflective film, and the surface shape was further increased to $1/25\lambda$. Based on this, a broadband HR coating from 750 to 1550 nm was formed on the surface of PMMA by adjusting the partial pressure of oxygen.

**Author Contributions:** Conceptualization: Z.L. and Q.L.; Methodology: Z.L.; Software: Q.L., X.Z., and C.S.; Validation: C.S., X.Q., X.W., and J.G.; Formal Analysis: Z.L.; Investigations: Z.L.; Resources: Z.L. and H.Y.; Data Curation: X.Z., C.S., and Q.L.; Writing—Original Draft Preparation: Z.L.; Writing—Review and Editing: Z.L., Q.L., and X.Q.; Visualization: Z.L.; Supervision: Z.L.; Project Administration: Z.L.; Funding Acquisition: Z.L.

**Funding:** This research was funded by the National Natural Science Foundation of China (Grant Nos. 61705226 and 61875193) and the Changchun Science and Technology Innovation "Shuangshi Project" Major Scientific and Technological Project (Grant No. 19SS004).

**Conflicts of Interest:** The authors declare no conflict of interest.

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
