# Peer review of "Broadband High-Reflection Dielectric PVD Coating with Low Stress and High Adhesion on PMMA"

_coatings, doi:10.3390/coatings9040237_

Round 1
Reviewer 1 Report
Interesting results and suggest minor revision
Use of long sentences followed by comma - better to avoid
few minor changes as below
1. Line 28: Delete on PMMA
2. Abbreviate AR and VR when first appear.
3. Line 64: Replace less thickness by thin
4.Sufix problem: TiO2, Ta2O5,ZrO2, HfO2 and Nb2O5 etc. Seen other places as well
5. Line: 99-100 In our previous work, AR coating on PMMA has been realized by e-beam evaporation. Cite you work here
6. Line 126: 1.7(HL)^5 use superscript
Author Response
Thanks so much for your meaningful comments. According to your comments, we have revised the manuscript point by point.
1.Use of long sentences followed by comma - better to avoid
We split some long sentences into separate concise sentences.
2.Line 28: Delete on PMMA
PMMA in line 28 has been deleted.
3. Abbreviate AR and VR when first appear.
AR and VR have been deleted and replaced by augmented reality and virtual reality, which can also be distinguished with antireflection (AR) coating.
4. Line 64: Replace less thickness by thin.
“less thickness” has been replaced by thin.
5. Sufix problem: TiO2, Ta2O5,ZrO2, HfO2 and Nb2O5 etc. Seen other places as well.
Sufix problem of “TiO2, Ta2O5,ZrO2, HfO2 and Nb2O5” in whole manuscript has been corrected.
6. Line: 99-100 In our previous work, AR coating on PMMA has been realized by e-beam evaporation. Cite you work here.
The expression in line 99-100 is not correct. We want to express the antireflection coating is the earliest realized process. So, “In our previous work” is replaced by “In the initial stages of the experiment”.
7. Line 126: 1.7(HL)^5 use superscript
Superscript problem has been corrected.
Reviewer 2 Report
Please find my comments below-
1. The manuscript needs serious reading for proper grammatical corrections. The manuscript should be rephrased and be written in more scientific and formal English as there are several poorly phrased sentences. Data is always spelled as ‘Date’ and single as ‘Shingle’.
2. Figure 2(b), the results are for PMMA+HC coating or PMMA+HC+AR coating? Please clarify.
3. There is no Figure 3(a)(b), it should be modified to Figure 4…(Lines 146- 162).
4. Line 161, what do author mean by repeatability error? Is it simply because of repeating measurement? Or if the same substrate is used for all the HC and AR coatings?
5. What are VR devices in abstract, I would suggest to rather write the full name.
6. Line 182, RMS of which parameter? Is it roughness?
7. Please put legend in Fig. 5a and Fig7b.
8. Lines 189- 193 (On the other hand………-165MPa), the results of different deposition rate should be presented separately. I believe they should not be part of environmental study.
9. Fig 7a, please use different markers so that plots can be readable in black and white print.
Thank you.
Author Response
First, thank you for the comments. According to your comments, we have revised the manuscript carefully.
1.The manuscript needs serious reading for proper grammatical corrections. The manuscript should be rephrased and be written in more scientific and formal English as there are several poorly phrased sentences. Data is always spelled as ‘Date’ and single as ‘Shingle’.
The manuscript has been revised carefully for grammatical corrections.
2.Figure 2(b), the results are for PMMA+HC coating or PMMA+HC+AR coating? Please clarify.
Figure 2(b), the measured transmittance results is only for PMMA+HC, and the photograph of three samples is mainly to illustrate that introducing the HC coating doesn’t influence the outlook and optical performance of coatings on PMMA.
3. There is no Figure 3(a)(b), it should be modified to Figure 4…(Lines 146- 162).
In Line 146-162, Figure 3(a) (b) has been modified to Figure 4(a) (b).
4.Line 161, what do author mean by repeatability error? Is it simply because of repeating measurement? Or if the same substrate is used for all the HC and AR coatings?
Line 161, the expression is improved and revised by “Besides, it should be noted that the coating stress result of different samples is a range but not a constant value in Fig. 4(a) and 4(b). This is mainly caused by the errors measured in multiple experiments and temperature difference of film formation area of different samples.”
5. What are VR devices in abstract, I would suggest to rather write the full name.
“AR and VR devices” has been replaced by full name.
6. Line 182, RMS of which parameter? Is it roughness?
Line 182, RMS means surface roughness parameter, and we have made the corresponding change.
7. Please put legend in Fig. 5a and Fig7b.
Legend has been put in Fig. 5(a) and Fig. 7(a).
8.Lines 189- 193 (On the other hand………-165MPa), the results of different deposition rate should be presented separately. I believe they should not be part of environmental study.
Usually, deposition rate will influence the inner microstructure and surface morphology, which are strongly related to the quality of durability. We originally placed the deposition rate test here to demonstrate that different deposition rates do not affect durability for our process. So, we add “Samples with different deposition rate also experienced environmental testing, and the deposition rate does not affect coating durability”.
9. Fig 7a, please use different markers so that plots can be readable in black and white print.
For Figure 7(a), we have changed the markers so that plots can be readable in black and white print.
Reviewer 3 Report
In this work by Li and co-workers the authors have described the preparation and the properties of a high-reflection coating deposited onto PMMA. Please consider the following comments to improve this work:
Abstract contains encrypted acronyms. Every time a new acronym is introduced its meaning should be given. Please correct it.
Line 28 - PMMA is duplicated.
I suggest more references to particular parts of the introduction. At present, you cite about 5 works to one sentence, but the rest remains unreferenced. Make it more homogeneous to support all the claims, not just the selected ones.
Please improve the English. An example of the part written in a rather peculiar way: "At present, coatings with less thickness could be successfully deposited onto PMMA. Less thickness means a lower probability of stress."
(Line 64 and 65) I suggest introduction at least some of te state of the art regarding coating of PMMA with the coatings that you have selected. At present your introduction suggests that it is nonexisting and you pioneer this area, which is probably not the case.
Please include absorbance spectrum of uncoated PMMA as well.
Caption to Figure 2 is wrong. (a) should be swapped with (b). Additionally about the plot, please be consistent and draw the first panel with the same style as Fig. 1 and Fig. 2b.
" The low refractive index material of HR coating is SiO2, while the high refractive index material will be selected from TiO2, Ta2O5,ZrO2, HfO2 and Nb2O5 through experimental comparison. " - could you please show me where are all the absorbance plots for these combinations? Ideally I would want to see this in a single overlaid plot to enable the reader to judge the effect of every coating on the transmittance.
Formula in Line 125 should be written properly as an equation.
"Deposition parameters are tried to be controlled the same." - please clarify this very worrying statement. What were the exact parameters for all the coating procedures? 11) Descirptions of axes on all the plots are barely visible. Insets in the plots are even worse - that makes it impossible to judge this manuscript properly.
Author Response
Thank you for kindly comments. According to your comments, we have revised the manuscript carefully.
1.Abstract contains encrypted acronyms. Every time a new acronym is introduced its meaning should be given. Please correct it.
Encrypted acronyms problem has been corrected.
2. Line 28 - PMMA is duplicated.
Duplicated word “PMMA” in Line 28 has been deleted.
3. I suggest more references to particular parts of the introduction. At present, you cite about 5 works to one sentence, but the rest remains unreferenced. Make it more homogeneous to support all the claims, not just the selected ones.
We think the cited references are enough to help understanding the introduction, while we have rearranged the location of some references and revised the issue of citations in large segments.
4. Please improve the English. An example of the part written in a rather peculiar way: "At present, coatings with less thickness could be successfully deposited onto PMMA. Less thickness means a lower probability of stress." (Line 64 and 65)
The English expression in Line 64 and 65 has been improved.
5. I suggest introduction at least some of the state of the art regarding coating of PMMA with the coatings that you have selected. At present your introduction suggests that it is nonexisting and you pioneer this area, which is probably not the case.
Your suggestion is very correct. Preparing coatings on PMMA is an issue that has been studied for a long time. As we explained in Line 51-54, there are a lot of deposition processes to realize coatings on PMMA, while physical vapor deposition (PVD) is the leading choice for mass production, low cost and productive efficiency. But it currently still faces a number of challenges, such as stress and cohesion, which are just the core research question.
6. Please include absorbance spectrum of uncoated PMMA as well.
According to your suggestion, transmittance spectrum of uncoated PMMA has been added into Figure 1. The absorbance of PMMA substrate in visible spectrum is less than 0.1%.
7. Caption to Figure 2 is wrong. (a) should be swapped with (b). Additionally about the plot, please be consistent and draw the first panel with the same style as Fig. 1 and Fig. 2b.
Figure 2 (a) has been swapped with (b). The Fig. 2(a) is obtained directly from K-space stress test instrument. It’s very difficult to change its plot form.
8. "The low refractive index material of HR coating is SiO2, while the high refractive index material will be selected from TiO2, Ta2O5,ZrO2, HfO2 and Nb2O5 through experimental comparison. " - could you please show me where are all the absorbance plots for these combinations? Ideally I would want to see this in a single overlaid plot to enable the reader to judge the effect of every coating on the transmittance.
For this comment, it’s our problem that we did not explain very well to the material absorbance. In this paper, coatings on PMMA including antireflection and high-reflection are mainly for visual spectrum. The absorbance of all mentioned coating materials at the normal deposition condition are very low and negligible when design and discuss the coating optical performance. And we add some explanation in Line 155. However, when we change the normal deposition parameters, especially the oxygen partial pressure, the absorbance of coating material cannot be ignored. As shown in Fig. 7(a), the extinction coefficient can represent the material absorption strength for optical thin film.
9. Formula in Line 125 should be written properly as an equation.
Formula in Line 125 is not a calculation equation. This formula means the structure of optical thin film, but we are sorry that the formula is not complete and revised to “PMMA Substrate / (HL)5 1.3(HL)5 1.7(HL)5 / Air”.
10. "Deposition parameters are tried to be controlled the same." - please clarify this very worrying statement. What were the exact parameters for all the coating procedures?
The wrong expression “Deposition parameters are tried to be controlled the same” has changed to “For processes of using different materials, deposition parameters were tried to keep consistent, including deposition rate, heating temperature, ion source power, sample spatial location, and oxygen partial pressure”. The exact parameters for all the coating procedures are showed after this expression. Evaporation rates are all 0.3 nm/s, background vacuum is 9.0×10-4 Pa, heating temperature is 220℃, voltage and current of RF ion source are 1100V and 1000mA, sample is 1.3m height above electronic gun, coating vacuum after oxygenation is 2.0×10-2 Pa, and coating thickness is 300nm.
11. Descriptions of axes on all the plots are barely visible. Insets in the plots are even worse - that makes it impossible to judge this manuscript properly.
Descriptions of axes on all the plots have been improved.
Reviewer 4 Report
In this manuscript, the authors study the deposition of layers onto PMMA substrates to improve its adhesion and properties. The manuscript is recommended for publication after the following revisions:
1. The abstract is not clear and should be improved. Please indicate which mean AR and VR devices and PMMA HR coating. Explain better the meaning of “…substrate by introducing a matching ratio of the connection layer.”
2. In line 28, please cut one PMMA.
3. In line 77, please indicate the size of the PMMA substrates and specially the thickness.
4. Is Figure 1 showing the spectrum of the coating or the one of coating on PMMA substrate? Please clarify.
5. In line 116, the thickness of the film attained should be indicated.
6. In line 125, the authors state “The thin film formula is very simple,…”. Please explain the meaning of thin film formula. Explain also how many layers will be considered.
7. In line 160, the authors refer to Figure 3a) and 3b) which do not exist in the paper. I suppose that the authors are referring figures 4 a9 and b). Please improve the introduction of these figure.
8. The manuscript should be read again carefully and the subjects better explained. The readers cannot be familiarized with this thematic.
Author Response
Thank you for your kindly comments. According to your comments, we have revised the manuscript point by point.
1. The manuscript is recommended for publication after the following revisions: The abstract is not clear and should be improved. Please indicate which mean AR and VR devices and PMMA HR coating. Explain better the meaning of “…substrate by introducing a matching ratio of the connection layer.”
The abstract is revised by fixing encrypted acronyms problem and changing some explanation sentences.
2. In line 28, please cut one PMMA.
In line 28, PMMA has been deleted.
3. In line 77, please indicate the size of the PMMA substrates and specially the thickness.
The size of the PMMA substrates is added in Line 82.
4. Is Figure 1 showing the spectrum of the coating or the one of coating on PMMA substrate? Please clarify.
Figure 1 has been updated. Now there are two transmittance curves in Fig.1. One is PMMA substrate, and the other one is PMMA substrate with two sides AR coating.
5. In line 116, the thickness of the film attained should be indicated.
The physical thickness of HC coating is almost 1.05μm, which can be calculated by transmittance curve or directly measured by stylus profiler. The thickness results obtained by two methods are nearly the same. And the thickness uniformity error is less than ±1% from the center to the edge of PMMA sample. We add the explanations in Line 123.
6. In line 125, the authors state “The thin film formula is very simple,…”. Please explain the meaning of thin film formula. Explain also how many layers will be considered.
We are sorry that the formula is not complete and clear. It has been revised to “Substrate / (HL)5 1.3(HL)5 1.7(HL)5 / Air”. There are all 30 layers of HR coating.
7. In line 160, the authors refer to Figure 3a) and 3b) which do not exist in the paper. I suppose that the authors are referring figures 4 a) and b). Please improve the introduction of these figures.
In Line 146-162, Figure 3(a) (b) has been modified to Figure 4(a) (b).
8. The manuscript should be read again carefully and the subjects better explained. The readers cannot be familiarized with this thematic.
By carefully revising, we have improved the expressions as far as possible.
Reviewer 5 Report
1. It will be useful to mention what is the PMMA transparency % for white light.
2. Please explain how does different material adhere to PMMA? Does each material have their own matching ratio?
3. Figure 1 needs more explanation on the caption. Each figure and caption should be self-explanatory.
4. Please correct Figure 2 caption. a and b are in reverse order.
5. Please make Figure 3 again.
Need to make better Figures and captions.
Author Response
Thank you for your meaningful comments.
1. It will be useful to mention what is the PMMA transparency % for white light.
The PMMA transparency for white light has been added in Figure 1.
2. Please explain how does different material adhere to PMMA? Does each material have their own matching ratio?
Different materials are all deposited onto the PMMA substrate by the same process, which is ion source assisted electron beam evaporation. We introduced the deposition method in details in Line 88-94. Nowadays, it is a typical physical vapor deposition method, especially good for high quality mass production.
3. Figure 1 needs more explanation on the caption. Each figure and caption should be self-explanatory.
Figure 1 has been updated. Now there are two transmittance curves in Fig.1. One is PMMA substrate, and the other one is PMMA substrate with two sides AR coating.
4. Please correct Figure 2 caption. a and b are in reverse order.
Caption of Figure 2 (a) has exchanged location with caption of Figure 2 (b).
5. Please make Figure 3 again.
Figure 3 has been remade and improved.1.
6. Need to make better Figures and captions.
All figures have some revisions and been improved.
Round 2
Reviewer 2 Report
I am not sure the revised manuscript is read carefully.
1. At line 122, data I still written as date. A simple CTRL+F key can find the misspelled words, not sure why authors are not paying attention.
2. I would suggest to keep all plots in same format, means please use same line/ marker pattern in all other plots (Fig 3, 5 7(b)).
Author Response
Thanks so much for your meaningful comments. According to your comments, we have revised the manuscript point by point.
1. At line 122, data I still written as date. A simple CTRL+F key can find the misspelled words, not sure why authors are not paying attention.
Sorry for our stupid mistake and our manuscript has careful revised by MDPI English editing service.
2. I would suggest to keep all plots in same format, means please use same line/ marker pattern in all other plots (Fig 3, 5 7(b)).
We have remade all plots and keep them in same format as far as possible except Fig. 2(a), which was directly obtained by the nano intender system and could not be revised.
Reviewer 3 Report
Although some comments have been taken into the consideration, which I am thankful for, many of them were fulfilled with the least possible effort and some errors remain in the revised version of the manuscript. Captions are not very informative and the level of English did not improve that much. Example from the previous round of review:
4. Please improve the English. An example of the part written in a rather peculiar way: "At present, coatings with less thickness could be successfully deposited onto PMMA. Less thickness means a lower probability of stress." (Line 64 and 65)
The English expression in Line 64 and 65 has been improved.
I gave you an example, so you fixed only this one problem rather than focus on improving the whole work.
Moreover, selection of colors in Fig. 6a is not very fortunate because both curves overlap and one cannot see the state before and after without paying extreme attention.
Author Response
First, thank you for the comments. According to your comments, we have revised the manuscript carefully.
1. Although some comments have been taken into the consideration, which I am thankful for, many of them were fulfilled with the least possible effort and some errors remain in the revised version of the manuscript. Captions are not very informative and the level of English did not improve that much.
All captions have been reedited and the English writing has been improved by MDPI English editing service.
2. Moreover, selection of colors in Fig. 6a is not very fortunate because both curves overlap and one cannot see the state before and after without paying extreme attention.
Fig. 6a has been improved. We changed the colors of these two curves to distinguish them. The two curves have a high degree of coincidence, which just indicate that the reflectance of the film changes little after environmental testing, and its reliability is very high.
Reviewer 5 Report
All figure captions MUST have complete explanation of what the figures mean and represent. Readers often see the Figures figures first and the figure caption must have a full description of what the figure is about.
Author Response
Thank you for kindly comments. According to your comments, we have revised the manuscript.
1. All figure captions MUST have complete explanation of what the figures mean and represent. Readers often see the Figures first and the figure caption must have a full description of what the figure is about.
All captions have been carefully reedited to explain the meaning of figures.